# Significant Predictors of Sports Performance in Elite Men Judo Athletes Based on Multidimensional Regression Models

**DOI:** 10.3390/ijerph17218192

**Published:** 2020-11-06

**Authors:** Maciej Kostrzewa, Radosław Laskowski, Michal Wilk, Wiesław Błach, Angelina Ignatjeva, Magdalena Nitychoruk

**Affiliations:** 1Institute of Sport Sciences, The Jerzy Kukuczka Academy of Physical Education, 40-065 Katowice, Poland; m.kostrzewa@awf.katowice.pl (M.K.); m.wilk@awf.katowice.pl (M.W.); angelina.ignatjeva09@gmail.com (A.I.); 2Department of Physiology and Biochemistry, Gdansk University of Physical Education and Sport, 80-336 Gdańsk, Poland; radoslaw.laskowski@awf.gda.pl; 3Department of Sport, University School of Physical Education, 51-612 Wrocław, Poland; wieslaw.blach@awf.wroc.pl

**Keywords:** judo, regression models, combat sports

## Abstract

Background: This research aimed to identify the most significant predictors of sports level using regression modeling. Methods: This study examined 16 judokas (aged 23 (±2.5)) from four weight categories, with four athletes in each category (66 kg, 73 kg, 81 kg and 90 kg). Each athlete was a member of the Polish National Team, an international master class (IM) or national master class (M). The tests were carried out twice (every two weeks) during the pre-competitive season in the morning, after a 10-min warm-up. The tests were performed according to the following protocol: Explosive Strength Lower Limbs (ExSLL) [W], Strength Endurance Lower Limbs (SELL) [%], Explosive Strength Upper Limbs (ExSUL) [W], Strength Endurance Upper Limbs (SEUL) [%]. The relationships between the dependent variable (ranking score) and the other analyzed variables (predictors) were estimated using the one-factor ridge regression analysis. Results: There were significant intergroup and intragroup differences in the results of explosive strength and strength endurance of the lower and upper limbs. The best predictors were identified using regression modeling: ExSLL, SELL, and SEUL. Conclusions: Increasing the value of these predictors by a unit should significantly affect the scores in the ranking table. Correlation analysis showed that all variables that are strongly correlated with the Polish Judo Association (PJA) ranking table scores may have an effect on the sports performance.

## 1. Introduction

Sports performance is a multifactorial trait resulting from the interplay of individual, environmental, and task characteristics. Due to its complex, dynamic, and multidimensional nature, understanding the performance variability among athletes requires the adoption of a holistic perspective, which considers the integration of the levels, interacting at different scales during the performance [1]. The nature of combat sport is to strike, throw or grapple with an opponent [2]. Judo is a predominantly anaerobic, intermittent combat sport [3]. The effects of recovery type after judo combat on blood lactate removal and on performance in an intermittent anaerobic task requiring strength are explored [4]. Differences in fat-free mass and muscle thickness at various sites relate to performance level among judo athletes’ quickness, balance, and explosiveness [5]. Strength, correct posture, effective movements, and balance are critical in judo. Due to the complexity of movements, judo can be considered as a tool to increase strength of postural muscles [6]. Sporting performance is a multi-factor phenomenon and is affected by motor [7,8], technical and tactical [9,10], and mental preparation [11]. Explosive strength and strength endurance are particularly important in judo, since these abilities are correlated with the sports level of elite athletes. Explosive strength determines the performance of dynamic movement activities, sudden body turns, jumps, kicks, or change of directions of movement. Furthermore, strength endurance allows for generating submaximum power repeatedly during the fight [12,13].

The research carried out among national and international athletes allows for the determination of this dependence and increases their substantive and indirect predictive value. Above all, the analysis of explosive strength and strength endurance of judo athletes at elite levels can improve the efficiency of the training process and thus the effectiveness of their technique [14]. A comprehensive process of a certain strength training consists of properly planned tempo of exercises, number of repetitions and sets, external load, rest intervals, type of contraction, the choice of exercises and their sequence [15,16,17,18]. The analysis of the technique of performing a resistance exercise, as well as many more details related to somatic and anthropometric changes during the athletes development makes sports training more and more sophisticated while the coaching staffs are supported by specialists from various fields of science, and the athlete becomes a source of valuable [19,20,21]. Sports forecasting is usually treated as a classification problem where one class (win, lose or draw) can be predicted [22]. A large number of functions can be collected in sports forecasts. Especially in judo, there are many factors that can be used to predict a sporting result. Therefore, our research aimed to identify the most important of the selected sport level predictors using regression modeling.

## 2. Materials and Methods

### 2.1. Participants

The research material included the results of tests in an elite male judo athlete, selected based on mixed sampling. At the time of the research, each athlete was a member of the Polish National Team, an international master class (IM) or national master class (M). The sports level of the athletes was determined based on their place in the ranking of the Polish Judo Association (PJA) [23]. The study examined 16 athletes from four weight categories, with 4 athletes in each category (66 kg, 73 kg, 81 kg and 90 kg). The descriptive characteristics of particular groups are presented in Table 1:

### 2.2. Procedures

Up to 24 h prior to testing, all volunteers were recommended to avoid strenuous exercise as well as resistance. They were informed about the research protocol and possible risks as well the benefits of the study, and then gave informed written consent to participate in the study. The respondents were allowed to withdraw from participation in the tests at any stage of the experiment. The research protocol was approved by the Bioethics Committee for Scientific Research at the Jerzy Kukuczka Academy of Physical Education in Katowice. The tests were conducted in the Strength and Power Laboratory of the Academy of Physical Education in Katowice on Keiser Power Rack and Keiser Squat A300 strength training devices.

### 2.3. Data Collection

Body weight and body composition were determined using the InBody 570 electric impedance analyzer. The selected strength tests concerned the lower and upper limbs. The tests were carried out twice (every two weeks) during the pre-competitive season in the morning, after a 10-min warm-up of 5 min of work on the cycle ergometer at low intensity followed by several strength exercises without load, involving the upper and lower body. The included bench press on the Keiser Power Rack hydraulic resistance device and squats on the Keiser Squat hydraulic resistance training system. The one repetition maximum test (1RM) on the Keiser Squat and Keiser Power Rack was determined based on the protocol for the number of repetitions used by Baechle et al. [24]. After a warm-up according to the protocol used by Saeterbakken et al. [25], the participants carried out several maximum repetitions so that the number covered a range of 3 to 8 maximum repetitions. Based on the formula, the value of 1RM was calculated and used to calculate the value of the load used by each athlete to perform tests of explosive strength and strength endurance [24]. To evaluate the explosive strength, the athletes tested performed two repetitions at maximum speed while maintaining the correct technique with a load of 50% 1RM. Strength endurance was examined during a 30 s test, with athletes performing 30 repetitions (1 repetition per second) with the frequency controlled by the metronome. In both tests, the participants performed exercises at maximum speed. The athletes were verbally motivated during each test. A 15-min rest interval for a full recovery was administered between the tests. The tests were performed according to the following protocol: Explosive Strength Lower Limbs (ExSLL) [W] (a test of 2 repetitions of squat below the right angle at maximum speed with a load of 50% 1RM), Strength Endurance Lower Limbs (SELL) [%] (a test of 30 repetitions of squat below the right angle, performed continuously without rest, with a load of 50% 1RM), Explosive Strength Upper Limbs (ExSUL) [W] (a test of 2 repetitions of the bench press at maximum speed with a load of 50% 1RM), and Strength Endurance Upper Limbs (SEUL) [%] (a test of 30 repetitions of the bench press, performed continuously without rest, with a load of 50% 1RM). In the case of SELL and SEUL variables, the percentage of power decrement index determined by the ratio of mean values between the 1st and 30th repetitions was used for the analysis.

### 2.4. Statistical Analysis

The most useful methods and tools of statistical analysis were applied and tested. The significance level was set at *p* < 0.05. The normality of distribution of the variables was verified using the Shapiro–Wilk test. The results of the tests clearly showed that the variables had a normal or close to normal distribution (*p* > 0.05).

All relationships between the variables studied were determined using correlation analysis and the Pearson correlation coefficient. If statistically significant correlations were found, further relationships were studied to determine the size of the effect of the independent on the dependent variable. The relationships between the dependent variable (ranking score) and the other analyzed variables (predictors) were estimated using the one-factor ridge regression analysis [26,27,28,29].

After simplification, the biometric regression model received the following (formula (1)):Y = F(x_1, …, x_k; a_1, …, a_p) + ε = F(x,a) + ε(1)
where: x_j–determinant variable (x = |x_1, …, x_k|^T^), a_j–parameter (a = |a_1, …, a_p|^T^), ε–random component (also known as random factor or measurement error).

In conclusion, a complementary analysis of statistical data was carried out using the Statistica software package (StatSoft Company, Statistica 12, Krakow, Krakow, Poland, version 2020)

## 3. Results

The results of correlation analyses between the variables of explosive strength and strength endurance of the lower and upper limbs and the ranking score (PJA) are presented in Table 2:

The results show that, in the group of judo athletes from the 66 kg category, fitness significant correlations between the ranking score (PJA) and all analyzed variables were observed. The ExSLL variable showed the strongest correlation. In the 73 kg category, significant correlations between the ranking score and the analyzed variables were also observed. However, ExSLL and ExSUL correlations with Y were not strong. The SEUL and SELL variables were most strongly correlated. In the 81 kg category, significant correlations were found between Y and ExSUL, SELL, and. ExSLL No significant correlations with the SEUL variable were observed. The SELL variable showed the strongest correlation. In the 90 kg category, strong and significant correlations were observed between Y and SELL and SEUL. However, the correlation found for the ExSLL variable was significant but not very strong. No significant correlations were observed for the ExSLL variable. The SEUL variable in this weight category was most strongly correlated (Table 2).

The biometric models designed for individual weight categories is shown in Table 3:

Multiple ridge regression analysis, advancing for the dependent variable Y1_J66_pt, allowed for the determination of the following form of regression function (Table 3) based on specific values of individual independent variables that were significant in the model (formula (2)):Y1_J66_pt = 201,967 + 61,611 × ExSLL + 19,694 × SELL(2)

This means that if the value of the ExSLL increases by a unit, the mean score (Y1_J66_pt) will increase by 61 points. If the value of the SELL increases by a unit, the average score (Y2_J66_pt) will increase by ca. 19 points (Table 3).

For the dependent variable Y2_J73_pt, the following form of regression function was obtained (formula (3)):Y2_J73_pt = 201.432 + 19.412 × SEUL + 9.911 × SELL(3)

This means that if the value of the SEUL increases by a unit, the mean score (Y2_J73_pt) will increase by 19 points. Furthermore, if the value of the SELL variable increases by a unit, the average score (Y2_J73_pt) will increase by ca. 10 points (Table 3).

For Y3_J81_pt, the following form of regression function was determined based on specific values of individual independent variables significant in the model (formula (4)):Y3_J81_pt = 210.784 + 41.193 × SELL(4)

This means that if the value of the SELL increases by a unit, the mean score (Y3_J81_pt) will increase by 41 points T (Table 3).

The same analysis, advancing for Y4_J90_pt, determined the following form of regression function (formula (5)):Y4_J90_pt = 291.052 + 46.528× SEUL + 44.030 × SELL(5)

This means that if the value of the SEUL increases by a unit, the mean score (Y4_J90_pt) will increase by 46 points. Furthermore, if the value of the SELL increases by a unit, the average score (Y4_J90_pt) will increase by ca. 44 points (Table 3).

A regression analysis was also carried out, modeled to determine the predictors for the entire group of athletes, taking into account the weight differences in relation to the ranking point values presented in Table 4.

For the dependent variable Y5-Total, the following form of regression function was determined based on specific values of individual independent variables that were significant in the model (formula (6)):Y5_Total = 708.329 + 12.636 × ExSLL + 10.164 × SEUL(6)

This means that if the value of the ExSLL variable increases by a unit, the mean score (Y5-Total) will increase by 12 points. Furthermore, if the value of the SEUL variable increases by a unit, the average score (Y5-Total) will increase by ca. 10 points (Table 4).

## 4. Discussion

The aim of this study was to find the relationship between the tested variables of explosive strength and strength endurance of the upper and lower limbs, as well the sports performance of judo athletes in four weight categories. As a result, the best predictors of the sports level in judo were determined. An innovative aspect of the research was the performance of all measurements using modern measuring pneumatic devices from Keiser (Keiser Squat and Keiser Power Rack), which ensure the performance of movements with simultaneous recording of power. These devices allowed for performing the movement at maximum speed, allowing for the measurement of several variables related to muscular strength and power.

The factors responsible for success in judo can be specific to each weight category and can represent a compromise between body weight and maximizing motor skills [30].

Regression analysis showed that in the 66kg category, the most significant predictor determined by the model was ExSLL, with its increase by a unit expected to lead to an increase in the score by 61 points on average. The second significant variable was SELL. This confirms the fact that the athletes in the lightweight category are fighting at a very high pace, using techniques from all groups of throwing and grappling, and are characterized by a high level of agility. Success is determined by a skillful approach to the opponent or performing a sequence of offensive actions, which is why the lower limbs are more critical in lightweight categories [31]. In the 73 kg weight category, the regression model determined two predictors (SEUL and SELL). The first one, with an increase by a unit, should increase the score in the (PJA) ranking by 19 points on average. Furthermore, the results of the analysis of variance proved that there was a statistically significant variation in strength endurance of the upper limbs. Higher values of the variables were achieved by athletes with fewer points in the ranking. Therefore, it can be assumed that these variables, and consequently the upper limbs, have a statistically lower impact on sports performance in judo than the lower limbs. This is confirmed by the publications by Franchini et al. [7,32] and Adam et al. [10], who, based on the results of their research, came to similar conclusions. At the same time, the model verified the results of correlation analyses in these two weight categories, confirming the obtained correlations of these two variables with scores according to (PJA). In the 81kg group of athletes, the analysis of the examined variables shows that the SELL test showed higher values of this variable to the higher-ranked competitors. This might suggest that the athletes with higher sports performance had better strength endurance because they were able to maintain the measured power for longer in the periods of time of the test. Interestingly, the values determining ExSUL in this weight category were higher in lower-ranked athletes. Therefore, it can be concluded that the results of the analysis obtained in the examined group of leading Polish judokas are similar to the results of research conducted by Franchini and his colleagues [7,32,33]. The lower limbs are more important for achieving high scores in judo than the upper limbs and this is particularly evident in the medium and lower weight categories. In this weight category, the regression analysis showed that the most significant predictive factor determined by the regression model was SELL. An increase in the SELL by a unit should result in an improvement of 41 points. Fighting in medium weight categories is characterized by high dynamics of performed actions and frequency of their repetition. In these categories, techniques from all groups of throws are used, with attacks starting mainly with the work of the lower limbs. The athletes move much more during the fight than in heavy categories and use their lower limbs for defensive purposes, so it is critical for them to develop explosive strength and strength endurance. In the 90kg group of competitors, the analysis of variance determined a statistically significant difference in SEUL. Much better results were achieved by the athletes with higher scores in the ranking. There were no statistically significant changes in the lower limb samples in this weight category. Due to the specific body build, athletes in heavy categories are characterized by a lower frequency of attacks [34]. The fight is more static, and the judo athletes focus on fighting for the grip. Therefore, the results of the analyses support the findings of other researchers that, in the heavyweight categories, SEUL and ExSUL have significantly influenced sports performance [7,32,33,35,36,37,38]. The regression analysis revealed that the most significant predictor was SEUL, with an increase by unit resulting in an improvement by 46 points. In these two groups, regression modeling positively verified the results of correlation analyses. In the heavier categories (90 kg and above), athletes are less likely to use techniques that require rapid body rotation and a low approach between the opponent’s legs. More often, foot techniques and hand throwing techniques are used when a dominant grip is developed. During the grappling on the ground in heavy categories, holding techniques are more often performed, with the strength of the upper limbs playing a critical role [34]. The Y-total variable was also analyzed together for all weight categories. The ExSLL variable turned out to be the most important predictor here, with its increase by a unit resulting in a 12-point increase in the ranking. This confirms the findings that the lower limbs are more important in judo [7,32,33]. It should be noted that only the variables studied were used to determine the predictors, and, based on them, the improvement of results in the (PJA) ranking list can be predicted depending on the improvement of mean test values of the predictors. However, it is known that the improvement in sports performance is also affected by other variables such as aerobic capacity [39] and anaerobic capacity [40,41], training experience, technique, tactics [42,43], the level of mental and physical fitness on a specific day, judge, injuries, diet, and psyche [43,44,45,46,47], which were not used for this analysis. In light of the need to improve the sports skill level, the subject of modeling should be extended to include more tests and more independent variables that were not examined in the present study. These include the variables resulting from biomechanical and physiological analysis [48,49,50] and tactical and technical analyses [33,51,52]. Optimal improvements in the sports performance can be achieved by using comprehensive solutions that allow for a comprehensive evaluation of the potential of athletes, taking into account social, pedagogical, medical, biological, and mental indicators. Such assumptions for the choice of variables for research will be the basis for further measurements and analyses conducted by the author to develop the widest possible base of variables describing judo as a sport. The results of the research and analyses contained in this study can be used for practical applications. They can be used as tools to support the training process and contribute to its optimization. They can also serve as a tool to monitor the explosive strength and strength endurance of the lower and upper limbs. They can also be used in the recruitment and selection for judo.

The main muscles of the correct posture are the lower limbs. In judo, posture is the most important consideration. Hence, we suggest paying particular attention to the strength of the muscles of the lower extremities in the training process, which is related to the health aspect of judo practitioners.

## 5. Conclusions

There were significant intergroup and intragroup differences in the results of explosive strength and strength endurance of the lower and upper limbs. The best predictors were identified using regression modeling: ExSLL, SELL, and SEUL. Increasing the value of these predictors by a unit should significantly affect the scores in the ranking table. Correlation analysis showed that all variables that are strongly correlated with the (PJA) ranking table scores may have an effect on the sports performance.

## Figures and Tables

**Table 1 ijerph-17-08192-t001:** Descriptive statistics of characteristics variables for the 4 weight categories of judokas (mean and (SD)).

Parameters	<66 kg	<73 kg	<81 kg	<90 kg
Age (years)	22 (±3)	22 (±2)	24 (±3)	25 (±2)
Body mass (kg)	67.2 (±1.5)	75.3 (±2.2)	82.7 (±2.5)	93.1 (±2.7)
Body height (cm)	171 (±2.3)	174 (±2.6)	181 (±3.1)	183 (±4.4)
BMI (Body Mass Index)	22.5 (± 1.6)	23.6 (±2.1)	24 (±1.7)	27 (±3.1)
SMM (kg)	34 (±1.4)	39 (±2.5)	45 (±2.1)	46.5 (±2.5)
ExSLL [W]	1734.5 (±16.2)	1712.0 (±28.7)	1679.2 (±46.8)	1711.7 (±13.4)
ExSUL [W]	617.5 (±21.2)	615.7 (±10.1)	665.2 (±14.3)	752.5 (±15.5)
SELL [%]	87.9 (±2.3)	89.7 (±2.9)	92.2 (±3.2)	91.3 (±6.4)
SEUL [%]	76.3 (±6.8)	76.8 (±7.5)	82.5 (±7.8)	80.6 (±8)

SMM, Skeletal Muscle Mass.

**Table 2 ijerph-17-08192-t002:** Correlations of explosive strength with variables of strength endurance of the lower and upper limbs in weight categories of 66, 73, 81 and 90 kg and the ranking score (PJA).

Variable	ExSLL	SELL	SEUL	ExSUL
CATEGORY: 66 kg				
Y–PT	0.893 *	0.762 *	0.741 *	0.713 *
CATEGORY: 73 kg				
Y–PT	0.424 *	0.658 *	0.664 *	0.595 *
CATEGORY: 81 kg				
Y–PT	0.636 *	0.888 *	0.389	0.640 *
CATEGORY: 90 kg				
Y–PT	0.443	0.815 *	0.848 *	0.531 *

* Statistically significant correlations with *p* < 0.05.

**Table 3 ijerph-17-08192-t003:** Biometric models of significant ranking predictors among the variables of explosive strength and strength endurance of the lower and upper limbs in particular weight categories (66, 73, 81 and 90 kg).

Variable	Beta	B	*p*
CATEGORY 66 kg-Y1_J66; R^2^ = 0.838			
constant		201.967	0.021
ExSLL	0.775	61.611	0.022
SELL	0.424	19.694	0.031
CATEGORY 73 kg-Y2_J73; R^2^ = 0.871			
constant		201.432	0.026
ExSLL	0.719	19.412	0.025
SELL	0.670	9.911	0.026
CATEGORY 81 kg-Y3_J81; R^2^ = 0.866			
constant		210.784	0.038
SELL	0.683	41.193	0.022
CATEGORY 90 kg-Y4_J90; R^2^ = 0.956			
constant		291.052	0.026
ExSLL	0.894	46.528	0.025
SELL	0.785	44.03	0.026

**Table 4 ijerph-17-08192-t004:** Summary of ridge regression for all participants (Y5-Total; R^2^ = 0.922).

Variable	Beta	B	*p*
constant		708.329	0.011
ExSLL	0.464	12.636	0.039
SEUL	0.387	10.164	0.041

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
