# Peer review of "Significant Predictors of Sports Performance in Elite Men Judo Athletes Based on Multidimensional Regression Models"

_ijerph, 2020, doi:10.3390/ijerph17218192_

Round 1
Reviewer 1 Report
Significant predictors of sports level in judo athletes based on multidimensional regression models
Page 1
Line 33:
variables that are strongly correlated with the (PJA) ranking table scores may have an effect on the
Can the authors clarify what (PJA) stands for?
Page 2
Line 59 :
The research material included the results of tests in an elite group of judokas, selected based on mixed sampling.
Can the authors clarify that the they selected only men
I that respect i will recommend to change the title of the paper to
“Significant predictors of sports performance in Elite men judo athletes based on multidimensional regression models”
Page 6
Line 226, 227, 228:
The authors stated:
In the heavier categories (90 kg and above), athletes are less likely to use techniques that require rapid body rotation and a low approach between theopponent's legs. More often, foot techniques and hand throwing techniques are used when a dominant grip is developed.
Will be very valuable to support this with the appropriate references
Page 7
Line 259,260:
The authors stated:
The main muscles of the correct posture are the lower limbs. In judo posture is the most important. Hence, we suggest paying particular attention to the strength of the muscles of the lower extremities in the training process. Which is related to the health aspect of judo practitioners.
I propose this paragraph to be moved at the discussion section
Page 6
Line 231 -232:
The authors stated:
The ExSLL variable turned out to be the most important predictor here, with its increase by a unit resulting in a 12-point increase in the ranking. This confirms the findings that the lower limbs are more important in judo [26-28].
Can they clarify if this paragraph is related to 90 kg and heavier categories?
Limitations
I propose the authors to add a paragraph on the Limitation of the study
Only males, not all categories included small number of participants.
Best Regards
Nikos. Malliaropoulos
MD, Dip &MSc in SEM, PhD, F.FSEM (UK).
Author Response
Page 1
Line 33:
PJA - Polish Judo Association
Page 2
Line 59
The research material included the results of tests in an elite man judo athletes
We change the title of the paper to
"Significant predictors of sports performance in Elite men judo athletes based on multidimensional regression models"
Page 6
Line 226, 227, 228:
Will be very valuable to support this with the appropriate references
We add references number 29.
Page 7
Line 259,260:
The main muscles of the correct posture are the lower limbs. In judo posture is the most important. Hence, we suggest paying particular attention to the strength of the muscles of the lower extremities in the training process. Which is related to the health aspect of judo practitioners.
This paragraph was moved at the discussion section.
Page 6
Line 231 -232:
The authors stated:
The ExSLL variable turned out to be the most important predictor here, with its increase by a unit resulting in a 12-point increase in the ranking. This confirms the findings that the lower limbs are more important in judo [26-28].
Can they clarify if this paragraph is related to 90 kg and heavier categories?
This paragraph is related to all weight categories. We change this paragraph to: Y-total variable was also analyzed together for all weight categories.
Limitations
We add a paragraph on the Limitation of the study
The limitation of the study was only males and all categories included small number of participants.
Reviewer 2 Report
As a reviewer I have the following remarks.
- Please re-consider to use the following affiliation list: The Jerzy Kukuczka Academy of Physical Education, Katowice, Poland. Department of Sports Theory; m.kostrzewa@awf.katowice.pl, m.wilk@awf.katowice.pl, angelina.ignatjeva09@gmail.com, magda.aneta.krawczyk@awf.katowice.pl and the subscript 1, for each of these authors. Also spelling: Department. We don’t need to repeat the same name of the Academy.
- Table 1 needs an explanation: mean, standard deviation – we don’t know what these values are. I assumed that only males?
- Line 116. Your equation – it will be good to explain what is your y, x, k=?, etc.
- Line 118: “the Statistica software package” – please specify its version and the company name (v. X.X, Company, year).
- Table 2 – the correlations are stat sig.? What kind of the correlations?
- You may consider to do this analysis as an repetitive measurement with group as a factor. Your subjects have correlated measuremnts.Thank you.
Author Response
Point 1:
Please re-consider to use the following affiliation list: The Jerzy Kukuczka Academy of Physical Education, Katowice, Poland. Department of Sports Theory; m.kostrzewa@awf.katowice.pl, m.wilk@awf.katowice.pl, angelina.ignatjeva09@gmail.com, magda.aneta.krawczyk@awf.katowice.pl and the subscript 1, for each of these authors. Also spelling: Department. We don’t need to repeat the same name of the Academy.
Response 1: We re-considered affiliation list. We corrected the spelling of the word Department.
Point 2:
Table 1 needs an explanation: mean, standard deviation – we don’t know what these values are. I assumed that only males?
Response 2:
These values were expressed as means (± standard deviation). Male only - we changed the title and added: "The research material included test results in the elite judo athletes."
Point 3: Line 116. Your equation – it will be good to explain what is your y, x, k=?, etc.
Response 3:
We changed equation to a simplified version and explained the individual components:
Y=F(x_1,…,x_k;a_1,…,a_p )+ε=F(x,a)+ε
were:
x_j – determinant variable (x=|x_1,…,x_k |^T)
a_j – parameter (a=|a_1,…,a_p |^T)
ε – random component (also known as random factor or measurement error)
Point 4:
Line 118: “the Statistica software package” – please specify its version and the company name (v. X.X, Company, year).
Response 4:
We add: (StatSoft Company, Statistica 12, Poland version 2020)
Point 5:
Table 2 – the correlations are stat sig.? What kind of the correlations?
Response 5:
Correlations that are statistically significant are marked in Table 2 with the symbol *
Point 6:
You may consider to do this analysis as an repetitive measurement with group as a factor. Your subjects have correlated measuremnts.Thank you.
Response 6:
We consider this analysis as an repetitive measurement with group as a factor. They can be used as tools to support the training process and contribute to its optimization. They can also serve as a tool to monitor the explosive strength and strength endurance of the lower and upper limbs. They can also be used in the recruitment and selection for judo.
Reviewer 3 Report
the idea of prediction of sports performance is always interesting. Considering combat sports, an area in which there are a lack of studies, becomes still more interesting. in this study, high level athletes were evaluated and the results are consistent with the protocol.
I have some suggestions for the authors:
Introduction, lines 38-56: please, reorganize this part; firstly, introduce the general idea of sport performance and then add judo; add more details about the methods of sports performance prediction, as well as the rationale for this study;
Methods: line 64, please delete “all members of the Polish national team”
Describe PBF in table 1 as well as the expression of data (mean, media, SEM, SD…)
Line 68-75: can you assume that non athlete did not train? I suggest to modify the text, e.g.: all volunteers were recommended to avoid strenuous exercise as well as resistance…
Add an experimental design figure, detailing the moments of evaluation and tests;
Results: this section is not well organized and presented; for example, the authors describe an equation and its meaning.
In this, I have returned to methods and did not find PJA scores. So, explain the terms of equation clearly, beta, B. SELL, exSLL; then, provide a table with equations and statistics, respective terms and relation to the table 3 data.
What is the main reason for table 4? Maybe you can change the name of terms, such as Y5-jtotal?
Finally, could you provided a single model for all volunteers and the stratify for weight category?
Author Response
Point 1:
Introduction, lines 38-56: please, reorganize this part; firstly, introduce the general idea of sport performance and then add judo; add more details about the methods of sports performance prediction, as well as the rationale for this study
Response 1:
We reorganized introduction. We added more details about the methods of sports performance prediction, as well as the rationale for this study.
Point 2:
Methods: line 64, please delete “all members of the Polish national team”
Response 2:
We deleted “all members of the Polish national team”.
Point 3:
Describe PBF in table 1 as well as the expression of data (mean, media, SEM, SD…)
Response 3:
We described PBF - (Body Fat %) and expression of data.
Point 4:
Line 68-75: can you assume that non athlete did not train? I suggest to modify the text, e.g.: all volunteers were recommended to avoid strenuous exercise as well as resistance…
Response 4:
We modify the text to: all volunteers were recommended to avoid strenuous exercise as well as resistance.
Point 5:
Add an experimental design figure, detailing the moments of evaluation and tests;
Response 5:
Point 6:
Results: this section is not well organized and presented; for example, the authors describe an equation and its meaning.
Response 6:
Point 7:
In this, I have returned to methods and did not find PJA scores. So, explain the terms of equation clearly, beta, B. SELL, exSLL; then, provide a table with equations and statistics, respective terms and relation to the table 3 data.
Response 7:
PJA - Polish Judo Association, we made corrections in abstract.
Point 8:
What is the main reason for table 4? Maybe you can change the name of terms, such as Y5-jtotal?
Resonse 8:
We wanted to provide a major predictor for all volunteers. We changed the name of terms.
Point 9:
Finally, could you provided a single model for all volunteers and the stratify for weight category?
Response 9:
We provided a single model for all volunteers in Table 4 and described below the Table 4.
Reviewer 4 Report
The authors sought to examine muscular fitness-related predictors of judo performance, quantified by participant level. Explosive strength of the lower extremities, endurance of the lower extremities, and endurance of the upper extremities were all identified as significant predictors of participant ranking, insinuating that both lower- and upper-body muscular fitness play a role in judo success. I found the rationale for the paper to be not very convincing and do not understand why the authors only evaluated muscular fitness as a predictor of performance when the authors even describe that VO2max measures were obtained in the methods section. Furthermore, the statistical analysis, which involved dividing an already modest number of participants (n=16) into subgroups and then running correlations and regressions with only four subjects is inappropriate. Finally, extensive editing of English language and style is required.
I have the following major comments:
Abstract
- Please provide greater quantitative description of participant baseline characteristics using means +/- sd. For example, age, height, etc. would be of interest. It might also be worthwhile to provide some indication as to the performance-level of the studies athletes. For example, where they regional, national, or international caliber competitors? This might affect interpretation of study findings.
- Quantitative findings from the regressions and corresponding p-values are needed when reporting results.
- The authors conclude that all variables are strongly correlated with Judo ranking, which is inconsistent with the results where explosive strength of the upper limbs was not mentioned (although no data were provided). Please clarify
- The meaning of “PJA” (line 33) should be provided before the abbreviation is used
Introduction
- The authors use of the terms “explosive strength” and “strength endurance” are, I believe, inappropriate. Please consider replacing with “power” and “endurance”, which are more well recognized variables of muscular contractile performance.
- I recommend breaking this into two paragraphs, starting at line 45 with the sentence “The research carried out among…”
- Greater description as to how these findings may be used by trainers and athletes to optimize training prescription for Judo athletes is warranted.
Materials and Methods
- Why were athletes of different weight classes assessed? Given the relatively modest sample size it seems it would have been most appropriate to restrict analyses to a single late class
- The term “somatic characteristic” is inappropriate. These are “Descriptive Characteristics”
- The “Data Collection” section should be broken down into subsections and greater description of the various procedures should be provided. For example, what protocol was used for the VO2max test and what criteria defined attainment of VO2max? Also, why was VO2max measured if this was not included in the analysis?
- Lines 105-106: the sentence starting “to optimize the results” is subjective and should be eliminated
- The alpha level (e.g., P < 0.05) should be indicated in your statistical anlysis section
Results
- It is inappropriate to run correlations and regression models among such a small number of subjects (n=4; Tables 2 & 3). Consider combining the weight classes and re-running statistics.
Discussion
- Lines 177-178: the authors state that “the best predictors of the sports level in judo were determined”. Considering only 4 variables of muscular fitness were assessed (lower and upper body power and endurance) this statement is inappropriate. Only muscular fitness was truly evaluated. Why did the authors not evaluate other possible physiological predictors of performance? For example: https://www.mdpi.com/2075-4663/8/7/92.
Author Response
Point 1:
Please provide greater quantitative description of participant baseline characteristics using means +/- sd. For example, age, height, etc. would be of interest. It might also be worthwhile to provide some indication as to the performance-level of the studies athletes. For example, where they regional, national, or international caliber competitors? This might affect interpretation of study findings.
Response 1:
We provided a more accurate quantitative description of the participant's basic characteristics using means +/- SD. We also provided some hints about the sports level of the respondents: Each athlete was a member of the Polish National Team, an international master class (IM) or national master class (M).
Point 2:
Quantitative findings from the regressions and corresponding p-values are needed when reporting results.
Response 2:
Quantitative findings from the regressions and corresponding p-values are presented in Table 3.
Point 3:
The authors conclude that all variables are strongly correlated with Judo ranking, which is inconsistent with the results where explosive strength of the upper limbs was not mentioned (although no data were provided). Please clarify
Response 3:
The results of correlation analyses between the variables of explosive strength and strength endurance of the lower and upper limbs and the ranking score (PJA) are presented in Table 2.
Point 4:
The meaning of “PJA” (line 33) should be provided before the abbreviation is used
Response 4:
We explained the meaning of PJA - Polish Judo Association.
Point 5:
The autrength endurance of the lower and upper limbs and the ranking score (PJA) are presented in Table 2.hors use of the terms “explosive strength” and “strength endurance” are, I believe, inappropriate. Please consider replacing with “power” and “endurance”, which are more well recognized variables of muscular contractile performance.
Response 5:
Explosive strength and strength endurance are particularly important in
judo, since these abilities are correlated with sports level of elite
athletes. Explosive strength determines the performance of dynamic
movement activities, sudden body turns, jumps, kicks, or change of
directions of movement. Furthermore, strength endurance allows for
generating submaximum power repeatedly during the fight [7,8]. Therefore
explosive strenght seems to be a more precise term for judo's motor
ability than power. Whereas endurance is a general term, it may refer
to, for example, aerobic endurance, in judo combat it seems to be more
precise term strenght endurance.
Point 6:
I recommend breaking this into two paragraphs, starting at line 45 with the sentence “The research carried out among…”
Response 6:
We broke this into two paragraphs.
Point 7:
Greater description as to how these findings may be used by trainers and athletes to optimize training prescription for Judo athletes is warranted.
Response 7:
We added greater description as to how these findings may be used by trainers and athletes to optimize training prescription for Judo athletes is warranted.
Point 8:
Why were athletes of different weight classes assessed? Given the relatively modest sample size it seems it would have been most appropriate to restrict analyses to a single late class
Resonse 8:
The analyses for all volounteers are in Tabe 4. We assessed various weight categories, because the specificity of sports fight in judo differs depending on the weight category, which was demonstrated by the differences between the groups of the conducted analyzes.
Point 9:
The term “somatic characteristic” is inappropriate. These are “Descriptive Characteristics”
Response 9:
We changed term “somatic characteristic” to “Descriptive Characteristics”
Point 10:
The “Data Collection” section should be broken down into subsections and greater description of the various procedures should be provided. For example, what protocol was used for the VO2max test and what criteria defined attainment of VO2max? Also, why was VO2max measured if this was not included in the analysis?
Response 10:
We removed VO2max because it was not included in the analysis.
Point 11:
Lines 105-106: the sentence starting “to optimize the results” is subjective and should be eliminated
Response 11:
We eliminated sentense “to optimize the results”. We changed it to: We used the most useful methods and tools of statistical analysis were applied and tested.
Point 12:
The alpha level (e.g., P < 0.05) should be indicated in your statistical anlysis section
Response 12:
The alpha level (eg, P <0.05) is indicated in the statistical analysis section: The significance level was set at p <0.05.
Point 13:
It is inappropriate to run correlations and regression models among such a small number of subjects (n=4; Tables 2 & 3). Consider combining the weight classes and re-running statistics.
Response 13:
We assessed various weight categories, because the specificity of sports fight in judo differs depending on the weight category, which was demonstrated by the differences between the groups of the conducted analyzes.
Point 14:
Lines 177-178: the authors state that “the best predictors of the sports level in judo were determined”. Considering only 4 variables of muscular fitness were assessed (lower and upper body power and endurance) this statement is inappropriate. Only muscular fitness was truly evaluated. Why did the authors not evaluate other possible physiological predictors of performance? For example: https://www.mdpi.com/2075-4663/8/7/92.
Response 14:
Line 177-178: We wanted the best predictors from the ones we analyzed in this paper, not the best overall.
Round 2
Reviewer 3 Report
Except the modification of terms in table 4, other all suggestions were ammended.
Author Response
We changed term of Table 4: Summary of ridge regression for all participants (Y5-Total; R^2 =0,922)
Reviewer 4 Report
Though I appreciate the authors time spent revising their work and responding to reviewers I have serious reservations about the scientific soundness of performing correlation and regression analyses on small subsets (n=4) of an already modest sample (n=16). In such a small sample a single subject could drive the entire relationship. For this reason, many sports science and physiology journals request the authors show individual data for studies of < 12 (or more) subjects. Moreover, a true analysis of significant physiological predictors of performance would include a more comprehensive physiological assessment. Based on the authors response to my last inquiry it sounds as though they have these data but are not interested in showing them, which is an inaccurate representation of the study.
Author Response
Dear reviewer, we appreciate your valuable substantive comments.
Referring to them, in Table 1, we have included individual measurement
results for each group of players. We decided to research a small group
due to the high sports level of the surveyed players, who at that time
belonged to the national team. The groups were homogeneous with slight
variations which did not distort the results.
Best regards.